# Dopamine D2 receptor antagonist counteracts hyperglycemia and insulin resistance in diet-induced obese male mice

**Dina I. Vázquez-Carrillo, Ana Luisa Ocampo-Ruiz, Arelí Báez-Meza, Gabriela Ramírez-Hernández, Elva Adán-Castro, José Fernando García-Rodrigo, José Luis Dena-Beltrán, Ericka A. de los Ríos, Magdalena Karina Sánchez-Martínez, María Georgina Ortiz¤, Gonzalo Martínez de la Escalera, Carmen Clapp, Yazmín Macotela** *

Instituto de Neurobiología, Universidad Nacional Autónoma de México (UNAM), Campus UNAM-Juriquilla, Querétaro, México

¤ Current address: División de Ciencias de la Salud, Universidad Anáhuac Querétaro, Querétaro, México
* macotelag@unam.mx

**Data Availability Statement:** All relevant data are within the manuscript and will be submitted as

## Abstract

Obesity leads to insulin resistance (IR) and type 2 diabetes. In humans, low levels of the hormone prolactin (PRL) correlate with IR, adipose tissue (AT) dysfunction, and increased prevalence of T2D. In obese rats, PRL treatment promotes insulin sensitivity and reduces visceral AT adipocyte hypertrophy. Here, we tested whether elevating PRL levels with the prokinetic and antipsychotic drug sulpiride, an antagonist of dopamine D2 receptors, improves metabolism in high fat diet (HFD)-induced obese male mice. Sulpiride treatment (30 days) reduced hyperglycemia, IR, and the serum and pancreatic levels of triglycerides in obese mice, reduced visceral and subcutaneous AT adipocyte hypertrophy, normalized markers of visceral AT function (PRL receptor, Glut4, insulin receptor and Hif-1α), and increased glycogen stores in skeletal muscle. However, the effects of sulpiride reducing hyperglycemia were also observed in obese prolactin receptor null mice. We conclude that sulpiride reduces obesity-induced hyperglycemia by mechanisms that are independent of prolactin/prolactin receptor activity. These findings support the therapeutic potential of sulpiride against metabolic dysfunction in obesity.

## Introduction

Obesity, the most prevalent non-communicable disease of our time, triggers metabolic alterations leading to type 2 diabetes (T2D) and cardiomyopathy—chronic disorders that diminish the quality of life [1]. Obesity is defined as the excessive accumulation of adipose tissue that is deleterious to health [2], although it is recognized that some people with obesity remain metabolically healthy [3]. Metabolically unhealthy obesity is mainly derived from white adipose tissue (WAT) dysfunction, involving adipocyte hypertrophy, inflammation, macrophage infiltration, hypoxia, fibrosis, and atrophied angiogenesis and adipogenesis [4]. In obesity conditions, visceral adipose tissue (VAT) is more likely to become dysfunctional than

Supporting Information files upon acceptance of the manuscript.

**Funding:** This study was supported by the National Council of Humanities, Science and Technology of Mexico (CONAHCYT -https://conahcyt.mx) grant 284771 to YM. The funders had no role in study design, data collection and analysis, decision to publish, or preparation of the manuscript.

**Competing interests:** The authors have declared that no competing interests exist.

subcutaneous adipose tissue (SAT) [5–7]. Excessive accumulation of VAT is associated with increased development and severity of diseases (cardiometabolic, gastrointestinal, COVID-19) [8–10], and reverting the dysfunctional state of WAT results in improved glucose homeostasis and systemic insulin sensitivity [11–14]. Therefore, targeting adipose tissue dysfunction is key for treatments and strategies against metabolic alterations in obesity.

The hormone prolactin (PRL), mainly known for promoting milk production during lactation, exerts more than 300 functions in mammals [15, 16], including the regulation of metabolic homeostasis [17, 18]. Excessively high circulating PRL levels, as seen in patients with prolactinomas, are associated with metabolic dysfunction; moderately high PRL levels, defined as HomeoFIT-PRL [19], are protective against metabolic diseases in rodents and humans [20–22]; and low PRL levels correlate with a higher prevalence of metabolic diseases, including T2D and non-alcoholic fatty liver disease [19, 23, 24]. Diet-induced obese male rats have low PRL levels, and treatment with PRL improves their systemic insulin sensitivity and reduces adipocyte hypertrophy in VAT [13]. Moreover, low PRL levels correlate with a higher prevalence of T2D, insulin resistance, and adipocyte hypertrophy in humans [25]. These observations have led to the hypothesis that medications causing hyperprolactinemia within HomeoFIT-PRL levels could be beneficial against obesity and its comorbidities.

Sulpiride is a highly selective antagonist of dopamine D2 receptors used as a second-generation antipsychotic agent [26]. It has the side effect of increasing circulating PRL levels by blocking D2 receptors in pituitary lactotrophs, thereby preventing dopamine inhibition of PRL expression and secretion [27]. The high doses (> 600 mg/day) of sulpiride required for antipsychotic use have unwanted metabolic effects, including weight gain and hyperglycemia [28]. However, sulpiride is used at low doses (< 50 mg/day) as a digestive adjuvant in gastroparesis with no secondary metabolic effects [29, 30]. In healthy rodents, a high sulpiride dose also induces insulin resistance, increases serum free fatty acids (FFA) and triglycerides (TG), and leads to fatty liver [31]; whereas at low doses, it is beneficial against colon inflammation in a rat colitis model [32]. Here, we report that sulpiride improves glucose homeostasis and insulin sensitivity, prevents lipotoxicity and adipose tissue dysfunction in obese mice, and that these effects are independent from the effects of prolactin on its cognate receptors.

## Materials and methods

### Animals

Male C57BL/6 mice were housed at 22˚C on a 12-h/12-h light/dark cycle with free access to food and water. Eight-week-old mice were fed a control diet (Lean) (Laboratory rodent Diet 5001, LabDiet, Richmond, IN) or an obesogenic diet (Obese) with 60% fat (Open Source Diet D12492; Research Diets, New Brunswick, NJ) for eight weeks. After four weeks on the control or obesogenic diet, mice were treated with a daily dose of 30 mg/kg of intraperitoneal (i.p.) sulpiride (Dogmatil, 70062, Sanofi-Aventis, Barcelona, España) for 30 days. Body weight and food consumption were measured every week. Following the same protocol, male 8-week-old C57BL/6 mice, wild type (Prlr+/+) or null for the PRL receptor gene (Prlr-/-) were fed an obesogenic diet for eight weeks and treated with sulpiride or vehicle solution for the last 30 days. After this period animals were sacrificed using $CO_2$ inhalation followed by decapitation. The colony of prolactin receptor null mice has been expanded and maintained for many generations in the vivarium of our Institute and was originally started from Prlr-/+ mice on a C57BL/6 background, obtained from The Jackson Laboratory (www.jax.org; strain:003142, B6.129P2-Prlr^tm1Cnp/J) [33]). All studies were approved by the Bioethics Committee of the Institute of Neurobiology of the National University of Mexico (UNAM) (Protocol numbers 075 and 033), which complies with the *Guide for the Care and Use of Laboratory Animals*

published by the US National Institutes of Health (Eighth Edition, National Academy Press, Washington, D.C.).

## Serum measurements

Mouse serum was obtained after the 30 days of sulpiride treatment. PRL was evaluated by ELISA as previously reported [34]; insulin was measured using the Ultra Sensitivity Mouse Insulin ELISA Kit (90080, Crystal Chem, Elk Grove Village, IL); TG were measured using a colorimetric assay kit (10010303–96, Cayman Chem, Ann Arbor, MI); serum FFA was measured using the FFA Quantitation Kit (MAK044-1KT, Sigma-Aldrich, Burlington, MA); serum glycerol was measured using the Glycerol Assay Kit (MAK117-1KT, Sigma-Aldrich), and glucose levels were measured in fasting (4 h without food) and postprandial conditions (2 h without food) using glucose test strips (Accu-Check Active, 07124112, Roche, Mannheim, Germany). Glucagon levels were evaluated using a Mouse Glucagon ELISA kit (81518, Crystal Chem, Elk Grove Village, IL).

## Glucose and insulin tolerance tests

Glucose tolerance tests (GTT) and insulin tolerance tests (ITT) were performed in mice at the fourth weeks of sulpiride treatment, respectively. Blood glucose levels were measured in tail vein samples before or 15, 30, 60, 90, 120 min after an i.p. injection of 2 g/kg of 50% dextrose (PiSA Pharmaceuticals, Mexico) for GTT using glucose test strips (Accu-Check Active, Roche, Mannheim, Germany), or 0.75 U/kg insulin (Humulin R, Eli Lilly, Indianapolis, IN) for ITT.

## Adipose tissue histology

Subcutaneous inguinal and epididymal (visceral) adipose tissues from mice were fixed in 10% formalin, and sections of 7 μm were stained with hematoxylin and eosin. Adipocyte size was determined by calculating the mean adipocyte area using ImageJ Fiji software with the Adiposoft plugin [35, 36] from 10 images per animal (Microscope Olympus BX60) of six mice per group. The relative adipocyte number was calculated by dividing the fat pad weight by the mean adipocyte area in that fat depot.

## Homeostatic model assessment of insulin resistance

The homeostatic model assessment of insulin resistance (HOMA-IR) was calculated in mice using the formula: HOMA-IR = fasting serum insulin (μUI/ml) × fasting serum glucose (mmol/L) / 22.5 [37].

## Liver and pancreas triglycerides

Samples of liver and pancreas were collected and stored at -80˚C, then 150–200 mg of each tissue was homogenized and used for triglyceride quantification with a Triglyceride Colorimetric Assay kit (10010303–96, Cayman Chem).

## Liver and skeletal muscle glycogen levels

Samples of liver and skeletal muscle (gastrocnemius) were collected from animals with 4 h fasting and stored at -80˚C. Homogenized tissues were used to measure glycogen content with a colorimetric assay kit (700480, Cayman Chem, Ann Arbor, MI).

## RT-qPCR

Adipose tissues were frozen in liquid nitrogen and stored at -80˚C. Total RNA was extracted using Trizol reagent (Invitrogen, Carlsbad, CA), and cDNA was synthesized using the high-capacity cDNA reverse transcription kit (Applied Biosystems, Warrington, UK). PCR products were detected and quantified in a Bio-Rad CFX96 Real-Time System (Bio-Rad Laboratories, Hercules, CA) using Maxima SYBR Green / ROX qPCR Master Mix (Thermo Scientific, Auburn, AL) in a final reaction volume of 10 μl containing template and 0.5 μM of each of the primer pairs for mouse *prolactin receptor* (*Prlr)* (Forward primer: ACACGCGCAGATCTCC TTACCA; reverse primer: CCCCTTCTTGCACAGCCACTT), *glucose transporter 4* (*Glut4)* (Forward primer: TGATTCTGCTGCCTTTCTGT; reverse primer: GGACATT GGACGCTCTCTCT), *insulin receptor* (*Insr)* (Forward primer: CAGAATGTGACAGAGTTTGATGGG; reverse primer: CGGGTCAATATCCACCACAGT), *hypoxia-inducible factor 1 alpha (Hif1a)* (Forward primer: ACAAGT CACCACAGGACAG; reverse primer: AGGGAGAAAATCAAGTCG), *glucose 6 phosphatase* (*G6Pase)* (Forward primer: TCGGAGACTGGTTCAACCTC; reverse primer: ACAGG TGACAGGGAACTGCT), *phosphoenolpyruvate carboxykinase* (*Pepck)* (Forward primer: CTAACTTGGCCA TGATGAACC; reverse primer: CTTCACTGAGGTGCCAGGAG) genes, and as housekeeping the TATA *box-binding protein* (*Tbp)* gene (Forward primer: ACCCTTCACCAAT GACTCCTATG; reverse primer: TGACTGCAGCAAATCGCTTGG).

## Indirect calorimetry measurement

Indirect calorimetry of mice was measured using the metabolic cages (Harvard Apparatus) from the Behavioral Analysis Unit at the Institute of Neurobiology, UNAM. Animals were housed at 22˚C on a 12 h/12 h light/dark cycle with free access to food and water. Respiratory quotient (RQ) and energy expenditure (EE) were measured for two days in the fourth week of treatment with sulpiride.

## Body composition measured by magnetic resonance imaging

Body composition was determined by measuring subcutaneous and abdominal fat and muscle from arms and legs using magnetic resonance imaging (MRI). The images were acquired using a 7.0T MRI scanner, with an echo planar imaging pulse sequence, heavily-weighted T2 images (TR = 2 s). A total of 52 slices were acquired with a slice thickness of 400 μm. Volumetric data were converted to units of mass (grams).

## Statistics

Statistical data analysis was performed using GraphPad Prism 9.3.0 for Windows (GraphPad Software, San Diego, CA, USA). Differences between two groups were evaluated by Student's t-test and differences between three or more groups by one-way ANOVA followed by Tukey's post-hoc comparison test.

# Results

## Sulpiride increases PRL levels without affecting body weight or caloric intake

To determine the lowest dose at which sulpiride achieves the maximal PRL level (below 100 μg/L, consistent with HomeoFIT-PRL levels), sulpiride was administered i.p. once a day for four days at doses of 10, 20, 30, or 40 mg/kg. All sulpiride doses increased serum PRL levels compared to the control group (Fig 1A) ($F = 95.04$, $p < 0.0001$). The dose that fulfilled the

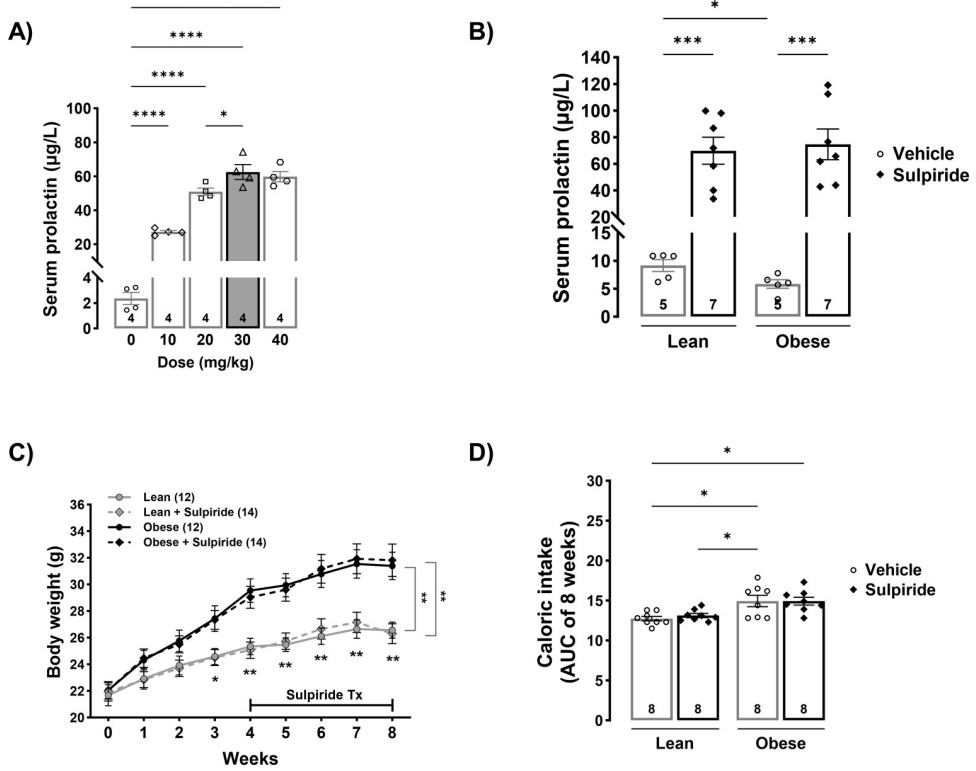

**Fig 1. Sulpiride increases serum PRL levels without affecting body weight or caloric intake.** A) Serum PRL levels after four days of sulpiride treatment administered i.p. once a day in C57BL/6 male mice. B) Serum PRL levels measured after 30 days of a daily i.p. dose of 30 mg/kg of sulpiride in lean and obese male mice. C) Body weight during the eight weeks of the experiment and 30 days of sulpiride treatment. D) Caloric intake from the 30 days of sulpiride treatment (number inside bars indicates the caloric intake per box of mice). White circles = vehicle treatment. Black rhombus = sulpiride treatment. * $p < 0.05$, ** $p < 0.01$, *** $p < 0.001$, *** $p < 0.0001$.

established criteria was 30 mg/kg, which increased PRL levels to 59.8 ± 2.7 μg/L (Fig 1A) and was subsequently used in all experiments. Next, male mice fed either a control or an obesogenic diet for eight weeks were injected daily with sulpiride or vehicle solution (saline) during the last 30 days of the eight-week period. Consistent with previous reports [13, 38], saline-injected obese mice showed lower ($p = 0.035$) PRL levels (5.8 ± 0.8 μg/L) than those fed a control diet (9.2 ± 1.1 μg/L) (Fig 1B). Sulpiride increased PRL levels to a similar extent in both groups (69.8 ± 10.1 in lean mice ($p < 0.001$) and 74.6 ± 11.5 in obese animals ($p < 0.0001$).

Sulpiride did not alter body weight or food intake in either lean or obese mice. During the eight weeks of the obesogenic diet feeding, mice treated with vehicle or sulpiride had an equivalent increase in body weight (42.6% and 44.7%, respectively). Similarly, mice fed a control diet increased their weight by 18.9% (vehicle treated) and 16.9% (sulpiride treated) (Fig 1C). These results were consistent with sulpiride having no effect on caloric intake in any of the diets (Fig 1D).

## Sulpiride improves glucose metabolism and insulin sensitivity in obese male mice

Next, we evaluated whether sulpiride influenced glucose and insulin tolerance as well as insulin sensitivity. According to the glucose tolerance tests, obese mice showed reduced glucose tolerance compared to lean mice, as evidenced by higher glucose levels during the test,

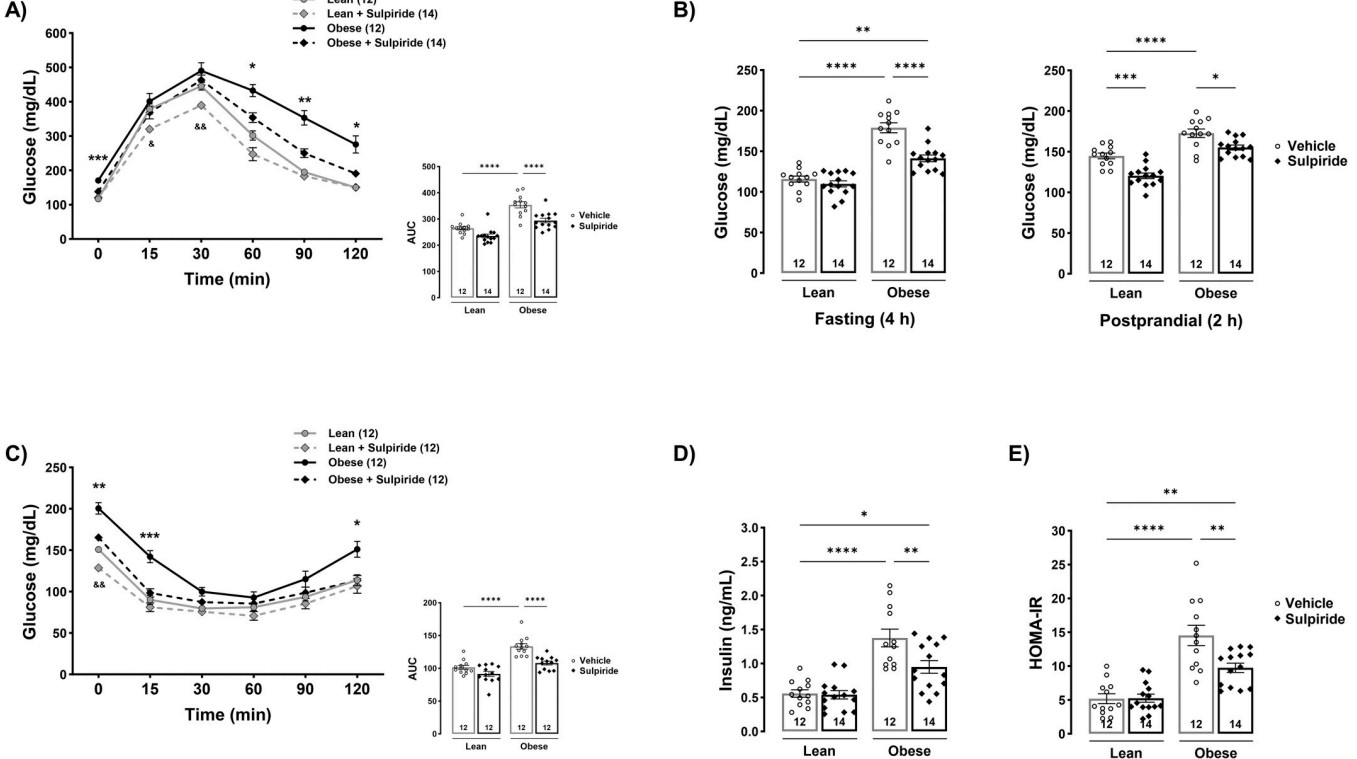

**Fig 2. Sulpiride counteracts hyperglycemia and insulin resistance in diet-induced obese male mice.** A) Glucose tolerance test and area under the curve performed after 4 h fasting in male mice treated with sulpiride for 30 days. B) Basal glucose levels in fasting (4 h) and postprandial (2 h) conditions after 30 days of sulpiride treatment. C) Insulin tolerance test and area under the curve performed after 2 h fasting and 30 days of sulpiride treatment. D) Serum insulin levels measured by ELISA and E) homeostatic metabolic assessment of insulin resistance (HOMA-IR) after 30 days of sulpiride treatment. White circles = vehicle treatment. Black rhombus = sulpiride treatment. * $p < 0.05$, ** $p < 0.01$, *** $p < 0.001$, **** $p < 0.0001$.

particularly at the later times of the experiments (minutes 60 ($p < 0.0001$), 90 ($p < 0.0001$) and 120 ($p < 0.001$)), while sulpiride-treated obese mice showed increased glucose tolerance (minutes 60 ($p < 0.01$), 90 ($p < 0.001$) and 120 ($p < 0.05$)), (Fig 2A). The effect of sulpiride on promoting glucose tolerance in obese mice was confirmed by the area under the curve (AUC) obtained from the GTT ($p < 0.01$), (Fig 2A). Sulpiride also lowered basal fasting glucose levels in lean mice ($p < 0.001$) and fasting ($p < 0.0001$) and postprandial ($p < 0.01$) glucose levels in obese mice (Fig 2B). To test whether sulpiride reduced glucose levels because of increased insulin secretion or improved insulin sensitivity, insulin levels and insulin tolerance were evaluated. Obese mice developed insulin resistance, as shown by a decreased response in reducing glucose levels upon insulin stimulation during the ITT (minutes 15 ($p < 0.001$), 30 ($p < 0.05$), 120 ($p < 0.05$)), while obese mice treated with sulpiride showed improved insulin sensitivity, evidenced by a larger decrease in glucose levels after insulin injection (minutes 15 ($p < 0.01$), 120 ($p < 0.05$); AUC ($p < 0.001$)) (Fig 2B). Regarding serum insulin levels, obese mice showed higher insulin levels, 1.54 ± 0.2 ng/mL compared to the lean mice, 0.57 ± 0.06 ng/mL ($p < 0.0001$), while sulpiride treatment in obese mice resulted in lower insulin levels, 1.15 ± 0.1 ng/mL ($p < 0.01$), (Fig 2C and 2D). This is consistent with the finding that sulpiride improves insulin sensitivity rather than increasing insulin levels. Furthermore, a 42% decrease in the HOMA-IR value was observed in obese mice treated with sulpiride compared to vehicle-treated obese mice ($p < 0.01$) (Fig 2E). Under chow diet conditions, sulpiride had no effect on insulin levels or HOMA-IR. Another possibility was that glucagon levels were decreased by

sulpiride, however, the levels of this hormone were not altered by sulpiride in either lean or obese mice (S1 Fig).

## Sulpiride prevents the accumulation of triglycerides in serum and pancreas of obese mice

To further evaluate the metabolic actions of sulpiride in obese mice, we analyzed the lipid profile of the animals. As expected, the obesogenic diet promoted a 36% increase in triglyceride serum levels (obese mice = 160 ± 12 mg/dL vs lean mice = 118 ± 3 mg/dL ($p < 0.001$)), a 50% increase in FFA (obese mice = 0.06 ± 0.002 nmol/uL vs lean mice = 0.04 ± 0.003 nmol/uL ($p < 0.01$)), a 63% increase in glycerol (obese mice = 3.1 ± 0.05 mg/dL vs lean mice = 1.9 ± 0.20 mg/dL ($p < 0.0001$)), and a 170% increase in total cholesterol (obese mice = 154 ± 8 mg/dL vs lean mice = 57 ± 2 mg/dL ($p < 0.0001$)) compared to values from lean mice (Fig 3A). Also, the obesogenic diet produced lipotoxicity, as evidenced by increased ectopic accumulation of TG in both liver (obese mice = 162 ± 8 mg/dL vs lean mice = 110 ± 11 mg/dL ($p < 0.001$)) and pancreas (obese mice = 67 ± 11 mg/dL vs lean mice = 17 ± 4 mg/dL ($p < 0.05$)) compared to mice on the control diet. Sulpiride treatment in obese mice decreased TG levels in serum (118 ± 5 mg/dL ($p < 0.001$)) and pancreas (38 ± 4 mg/dL ($p < 0.05$)), but had no effect on serum levels of FFA, glycerol, total cholesterol, or liver TG (Fig 3A–3C). Also, sulpiride had no effect on any of the lipid parameters evaluated in lean mice, neither on liver and pancreas weight in lean or obese mice (Fig 3B and 3C).

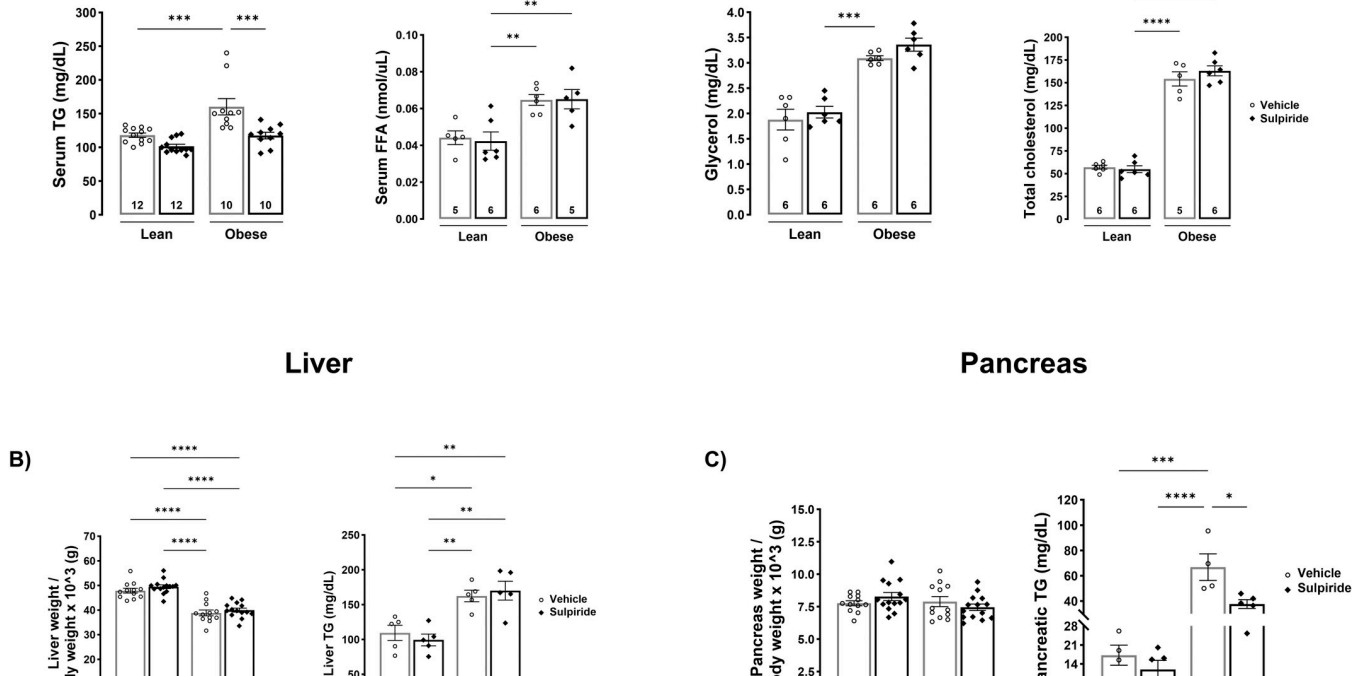

**Fig 3. Sulpiride decreased serum and pancreatic triglycerides levels in diet-induced obese mice.** A) Serum TG, FFA, glycerol and total cholesterol measured in male mice treated for 30 days with sulpiride or vehicle. B) Liver weight and TG measured by colorimetric assay after 30 days of sulpiride treatment. C) Pancreatic weight and TG measured by colorimetric assay after 30 days of sulpiride treatment. White circles = vehicle treatment. Black rhombus = sulpiride treatment. * $p < 0.05$, ** $p < 0.01$, *** $p < 0.001$, **** $p < 0.0001$.

## Sulpiride decreases adipocyte hypertrophy in visceral adipose tissue and prevents alterations in gene expression of markers of adipose tissue function in obese male mice

Since PRL treatment in obese rats reduces adipocyte hypertrophy [13], we evaluated whether sulpiride effects could be associated with an improved adipose tissue profile. As expected, in obese mice the ratio between visceral adipose tissue weight and body weight increased 2.5-fold compared to lean mice ($p < 0.0001$), and sulpiride did not affect this parameter in animals on either a control or an obesogenic diet (Fig 4A). The obesogenic diet did not alter the relative number of adipocytes, although there was a trend towards higher adipocyte numbers in the visceral AT from obese mice, and sulpiride treatment resulted in a significant increase in adipocyte numbers in obese mice compared to vehicle-treated lean mice ($p < 0.05$) (Fig 4B). Conversely, adipocyte size increased more than fourfold in obese mice compared to lean mice ($p < 0.0001$), which is consistent with adipocyte hypertrophy, while sulpiride decreased the adipocyte area by 39.6% in obese mice ($p < 0.001$), (Fig 4C). Moreover, adipose tissue from obese mice showed alterations in the gene expression of glucose and insulin metabolism genes. VAT from obese mice showed decreased expression of *Insr*, *Glut4*, and increased *Hif1a*, which is consistent with insulin resistance and dysfunctional adipose tissue [5, 10, 11], whereas sulpiride treatment counteracted these alterations (Fig 4D). Interestingly, *Prlr* expression was also decreased in the adipose tissue of obese mice, and sulpiride normalized its expression to levels compared to those seen in lean animals (Fig 4D). This result agrees with the finding that VAT from insulin-resistant subjects shows lower *Prlr* expression than that from insulin-sensitive ones [25].

## Sulpiride decreases adipocyte hypertrophy in subcutaneous adipose tissue without alterations in gene expression of markers of adipose tissue function in obese male mice

In subcutaneous adipose tissue, the obesogenic diet generated a 2.2 times increase in tissue weight compared to lean mice ($p < 0.0001$) without effects of sulpiride on any of the diets (Fig 5A). The relative number of adipocytes on SAT was increased in the obese mice ($p < 0.05$) and in the obese mice treated with sulpiride ($p < 0.0001$) in comparison with the lean mice, suggesting that the increase in the tissue weight is due to an expansion by hyperplasia which is a mechanism derived from adipogenesis and considered a healthy expansion of the adipose tissue [3]. Sulpiride increased the number of cells in the SAT of lean mice ($p < 0.05$) and there was a trend towards higher adipocyte numbers in the obese mice ($p < 0.0655$) (Fig 5B). In order to test whether this expansion was derived from an increase in the cell number and not by hypertrophy we evaluated the size of the adipocytes and observed that there was an increase of 2.3 times in the size of the adipocytes in the obese compared to the lean mice ($p < 0.001$) supporting that the expansion of the SAT in the obese mice was due to both hyperplasia and hypertrophy mechanisms. Sulpiride reduced by 33% the hypertrophy of the SAT from obese mice ($p < 0.05$), similar to the effect observed in the VAT (Fig 5C). Regarding gene expression related with adipose tissue functionality, SAT from obese mice showed decreased expression of *Glut4* without alterations on *Prlr*, *Insr* or *Hif1α*. This result agrees with the finding that SAT from obese subjects show lower GLUT4 expression than that from lean ones [25].

## Sulpiride does not affect fat and muscle composition evaluated by Magnetic Resonance Imaging (MRI)

To evaluate whether sulpiride has any effect on body composition (for instance increasing muscle mass to explain the improved glucose metabolism), an MRI was performed to measure

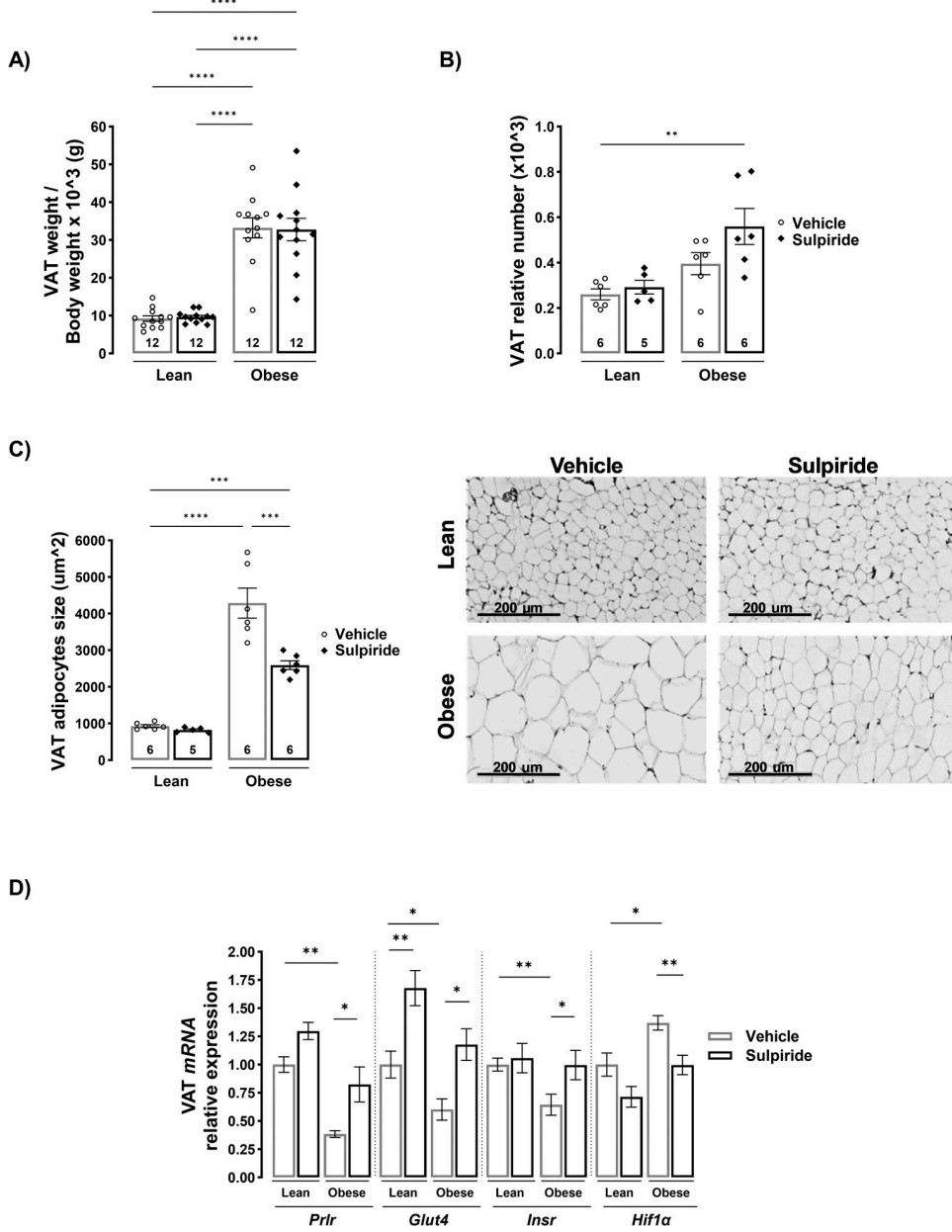

**Fig 4. Sulpiride reduces adipocyte hypertrophy and normalizes the gene expression of markers of visceral adipose tissue functionality in diet-induced obese male mice.** A) Visceral adipose tissue weight normalized to body weight from male mice treated for 30 days with sulpiride. B) Relative adipocyte number. C) Adipocyte size quantified in hematoxylin and eosin-stained tissue sections. D) Relative expression of genes involved in insulin sensitivity and hypoxia in visceral adipose tissue after 30 days of sulpiride treatment. Sample size (n) = 6–14 mice in each group. White circles = vehicle treatment. Black rhombus = sulpiride treatment. * $p < 0.05$, ** $p < 0.01$, *** $p < 0.001$, **** $p < 0.0001$.

the fat and muscle mass composition of the mice. While the obesogenic diet increased the VAT and SAT mass in mice, sulpiride did not change the fat mass in any of the diets (S2A Fig and S2B Fig). In foreleg and hind leg muscles, neither diet nor sulpiride affected the muscle tissue mass (S2C Fig1).

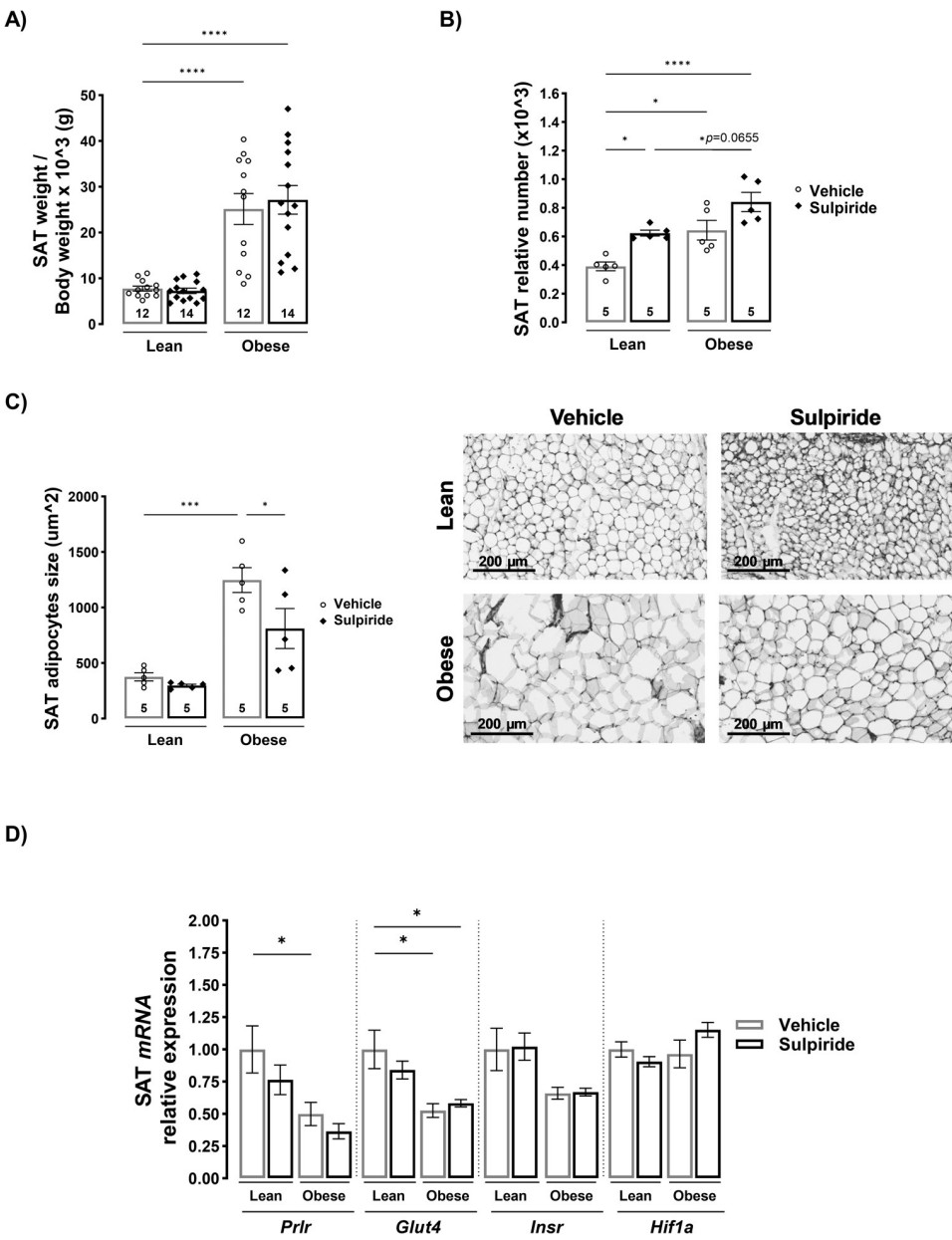

**Fig 5. Sulpiride reduces adipocyte hypertrophy without altering the gene expression of markers of subcutaneous adipose tissue functionality in diet-induced obese male mice.** A) Subcutaneous adipose tissue weight normalized to body weight from male mice treated for 30 days with sulpiride. B) Relative adipocyte number. C) Adipocyte size quantified in hematoxylin and eosin-stained tissue sections. D) Relative expression of genes involved in insulin sensitivity and hypoxia in subcutaneous adipose tissue after 30 days of sulpiride treatment. Sample size (n) = 5–6 mice in each group. White circles = vehicle treatment. Black rhombus = sulpiride treatment. * $p < 0.05$, ** $p < 0.01$, *** $p < 0.001$, **** $p < 0.0001$.

## Sulpiride does not alter the respiratory quotient or energy expenditure

Another possibility was that the effects of sulpiride reducing glucose levels in obese mice were due to increased energy expenditure. To test this, the respiratory quotient and the energy expenditure were evaluated in metabolic cages. The obesogenic diet decreased the respiratory quotient in obese mice during the dark cycle, indicating the use of other energy substrates

besides carbohydrates as primary energy sources, such as fat. Sulpiride did not affect the respiratory quotient in lean or obese mice. Regarding energy expenditure, there were no changes promoted by the diets or by sulpiride in any of the light/dark cycles (S3 Fig).

## Sulpiride increases glycogen content in skeletal muscle

To explore further the possible mechanisms by which sulpiride reduces glucose levels in obese mice, we evaluated the levels of glucose stored as glycogen on its main storage organs, liver and muscle, from lean and obese mice treated with and without sulpiride. Also, the gene expression of factors involved in hepatic gluconeogenesis were evaluated. Glycogen levels were decreased by the obesogenic diet in the liver and were not altered by the treatment with sulpiride in any of the dietary conditions (Fig 6A). On the other hand, muscle glycogen was not altered by the obesogenic diet, but was increased by sulpiride treatment in obese mice (Fig 6B), indicating that sulpiride promotes the storage of glucose in the skeletal muscle. Regarding liver gluconeogenesis, sulpiride had no effect on the gene expression of the main enzymes involved in this process: Glucose-6-phosphatase and Phosphoenolpyruvate carboxykinase 1 (Fig 6C).

## The effects of sulpiride reducing glucose levels are independent from prolactin receptor actions

Finally, to elucidate whether the effects of sulpiride reducing glucose levels in obese mice were mediated by enhanced PRL actions derived from the sulpiride-induced elevated PRL levels, we fed PRL receptor null mice (Prlr-/) and their wild type counterparts (Prlr+/+) with an obesogenic diet for 8 weeks and treated them with sulpiride or vehicle solution during the last 30 days of the 8-week period. Interestingly and contrary to our hypothesis, we found that sulpiride exerted its glucose-reducing action in both Prlr+/+ and Prlr-/- mice (Fig 7). Sulpiride improved glucose tolerance (Fig 7A), reduced fasting and postprandial glucose levels (Fig 7B and 7C), and increased insulin tolerance in both Prlr+/+ and Prlr-/- obese mice (Fig 7D). These results indicate that the actions of prolactin on its canonical receptors are not necessary for the effects of sulpiride to reduce glucose levels in obesity conditions.

## Discussion

Metabolic diseases such as T2D, insulin resistance, and adipose tissue dysfunction course with low systemic PRL levels [18, 19, 20, 21, 22, 23], whereas PRL administration improves insulin sensitivity and decreases adipocyte hypertrophy in HFD-induced obese male rats [13]. In this work, we tested whether sulpiride, a dopamine D2 receptor antagonist used in the clinic as an antipsychotic and prokinetic that elevates PRL levels, has metabolically beneficial actions in diet-induced obese male mice. Sulpiride treatment increased PRL levels, improved glucose tolerance and insulin sensitivity, lowered serum glucose, insulin, and triglyceride levels, and reduced pancreatic lipid accumulation in obese animals. These effects were independent of changes in body weight and caloric intake. Furthermore, in white VAT and SAT, sulpiride decreased adipocyte expansion due to hypertrophy, and in the VAT prevented the expression loss of *Prlr* and insulin sensitivity markers *Insr* and *Glut4*, and decreased *Hif1α* (hypoxia marker) expression in obesity conditions. However, contrary to our hypothesis, PRL actions were not involved in the effects of sulpiride reducing hyperglycemia in obese mice, as the drug exerted its glucose-reducing actions in mice lacking prolactin receptors. Therefore, sulpiride is metabolically beneficial in diet-induced obese male mice by mechanisms that are at least in part independent from PRL actions.

Chronic treatment with sulpiride increased PRL circulating levels (to around 60 μg/L) in lean and obese mice, a range within the so-called HomeoFIT-PRL values (> 25 μg/L

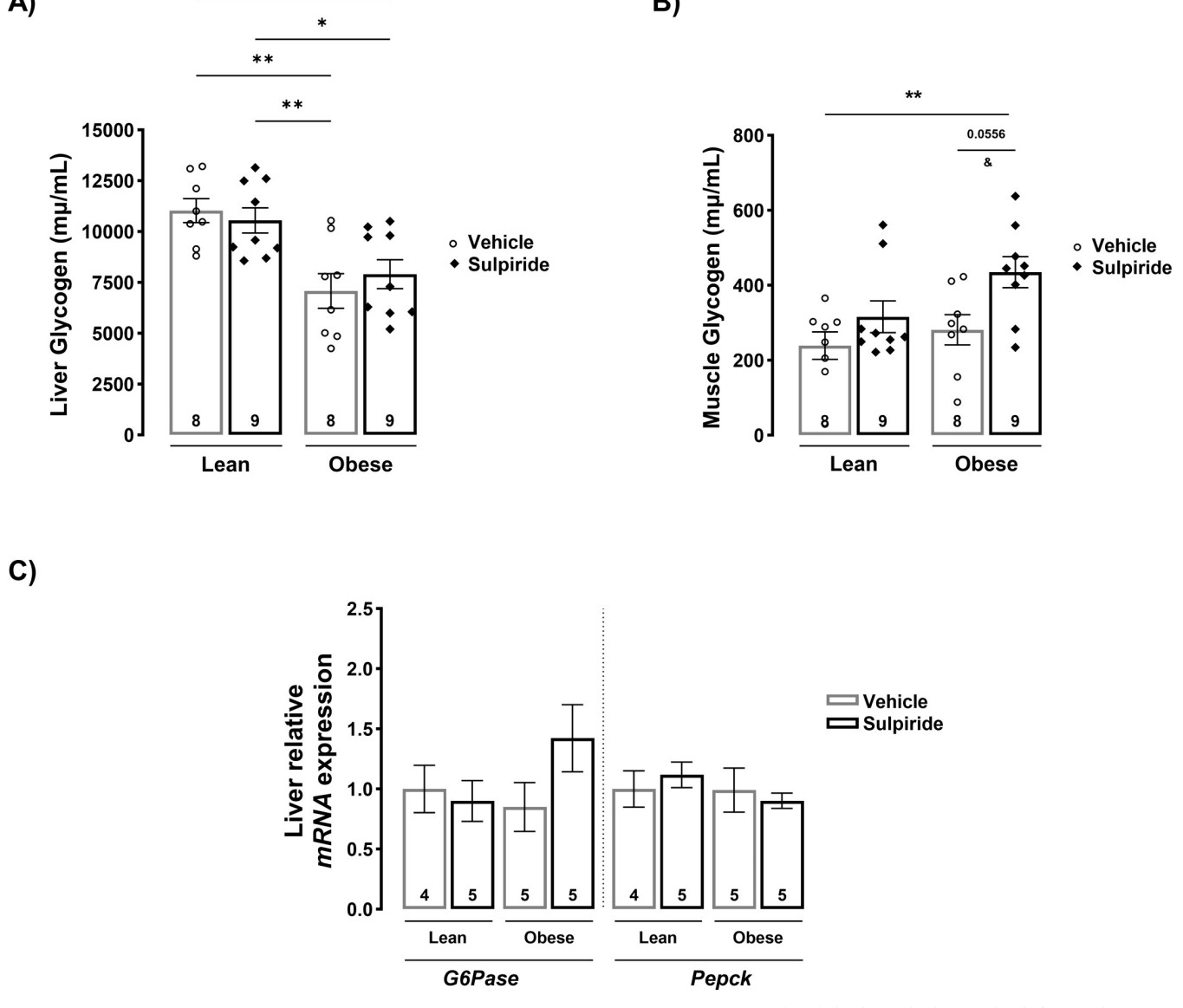

**Fig 6. Sulpiride increases skeletal muscle glycogen levels in diet-induced obese male mice.** A) Liver and B) skeletal muscle glycogen levels from male mice treated for 30 days with sulpiride. C) Relative expression of genes involved in gluconeogenesis in liver after 30 days of sulpiride treatment. White circles = vehicle treatment. Black rhombus = sulpiride treatment. * $p < 0.05$, ** $p < 0.01$. Besides one-way ANOVA and Tukey´s post-hoc comparison test, a T-test was performed in B) to compare glycogen levels between obese mice treated with or without sulpiride. ANOVA p = 0.0556, T-test [&] p = 0.01.

and < 100 μg/L) considered beneficial for metabolic homeostasis [18, 19]. Consistent with previous reports in rats, diet-induced obese mice had lower levels of PRL than lean mice [13, 38, 39], which indicates that an imbalance in PRL levels is associated with disrupted metabolic homeostasis under obesity. Of note, the finding that mice on an obesogenic diet had a similar elevation in serum PRL levels in response to sulpiride treatment compared to mice on a control diet suggests that reduced PRL levels in obese animals do not derive from decreased PRL secretory capacity [40–42].

Sulpiride improved glucose tolerance and reduced glycemia independently of alterations in body weight and caloric intake in both lean and obese mice. Even if these data are consistent with the beneficial metabolic actions of prolactin [13, 43], here we demonstrated that the

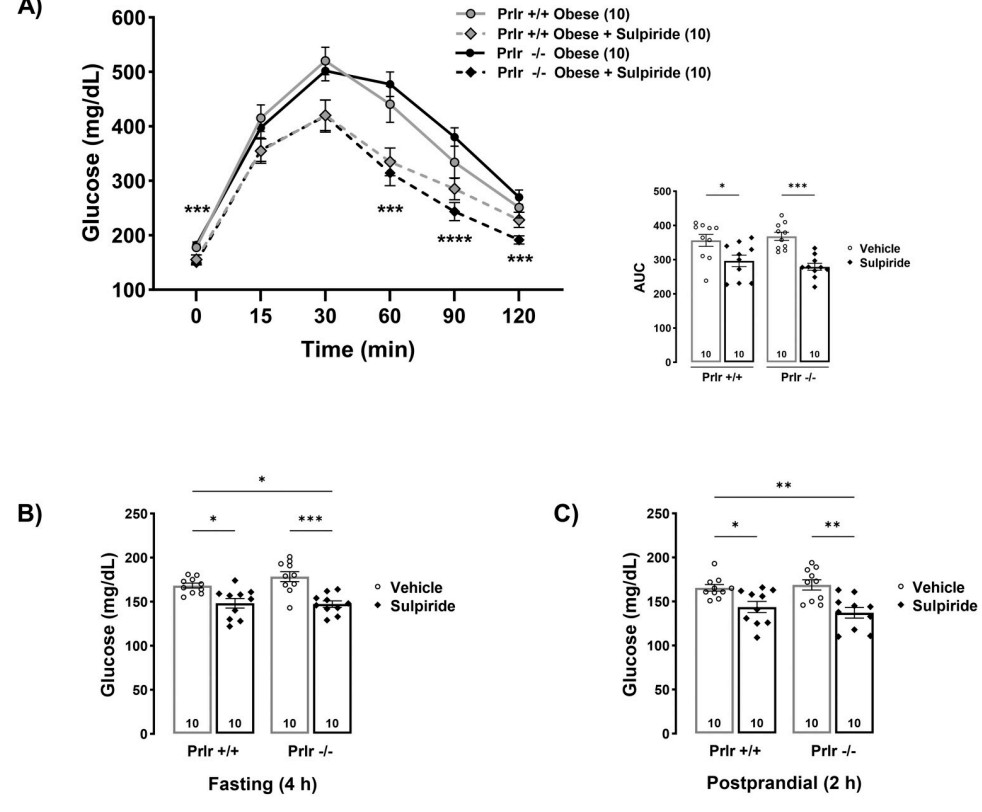

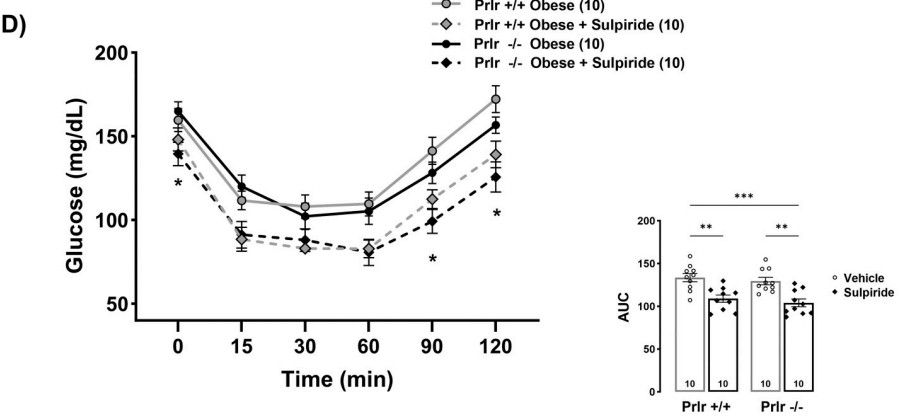

**Fig 7. Sulpiride reduces hyperglycemia, improves glucose tolerance and insulin sensitivity in diet-induced obese prolactin receptor null male mice.** A) Glucose tolerance test and area under the curve performed after 4 h fasting in male mice treated with sulpiride for 30 days. B) Basal glucose levels in fasting (4 h) and C) postprandial (2 h) conditions after 30 days of sulpiride treatment in diet-induced obese mice. D) Insulin tolerance test and area under the curve performed after 2 h fasting and 30 days of sulpiride treatment. White circles = vehicle treatment. Black rhombus = sulpiride treatment. * $p < 0.05$, ** $p < 0.01$, *** $p < 0.001$, **** $p < 0.0001$.

effects of sulpiride, at least reducing hyperglycemia are independent from PRL actions on its classical receptors, this was concluded because sulpiride reduced glucose levels and improved glucose tolerance in obese mice that are knockout for the prolactin receptors.

Sulpiride lowered glucose levels in obese hyperglycemic mice in both postprandial and fasting conditions. In contrast, in lean mice, sulpiride reduced glucose levels under postprandial but not fasting conditions, where glucose levels are already at a low basal level, suggesting that sulpiride enhances glucose uptake in conditions of elevated glucose levels (postprandial or HFD-induced). The finding that sulpiride did not reduce glucose levels in lean fasting mice supports the idea that sulpiride can be used to reduce hyperglycemia in obesity conditions without the risk of causing hypoglycemia. To explain sulpiride-induced reduced hyperglycemia, it is possible that surplus dopamine—generated by sulpiride occupation of dopamine D2 receptors—elicits a decrease in glucose levels, as reports have shown that administering a high dose of dopamine decreases systemic glucose levels in rats [44] and that dopaminergic tone is reduced in obesity [45–47]. Treatment with dopamine agonists has been shown to decrease [44, 48, 49] and increase [50] glucose levels. Of note, the mode or site of sulpiride delivery seems to be important for the metabolic outcome, as i.p. (systemic) injection of sulpiride results in reduced glucose levels in diet-induced obese animals (present study), whereas i.c.v. (central) injection of L-sulpiride leads to hyperglycemia [50]. It is suggested that inhibition of dopamine D2 receptors in the central nervous system increases glucose levels by elevating hepatic glucose production via sympathetic nerve activation [50], whereas low doses of sulpiride injected systemically do not cross the blood–brain barrier [51].

Most studies are consistent with our finding that sulpiride does not affect body weight gain in rodents or humans [52–57]. To the best of our knowledge, this is the first study where sulpiride is evaluated as a therapeutic agent against metabolic dysfunction derived from obesity. Baptista et al. reported increased glucose intolerance and hyperinsulinemia in male Wistar rats with no changes in body weight in response to sulpiride [52]. The differences with our study are that they used sulpiride together with an obesogenic diet as a model of drug-induced obesity, seeking to exacerbate obesity-derived alterations [52], while we approached the use of sulpiride as a possible therapeutic agent once obesity and metabolic alterations had already developed four weeks after the start of the HFD. In that study, a lower dose of sulpiride was used (20 mg/kg), and PRL circulating levels were not reported [52]. Of note, sulpiride exerts a stronger effect on elevating PRL levels in rats than in mice [53]. In male rats, a 20 mg/kg dose elevated PRL levels to 200 μg/L, whereas in our study with male mice, a 30 mg/kg dose of sulpiride elevated PRL levels to 60 μg/L. Therefore, it is likely that the high PRL level, together with the metabolic challenge of an HFD, has deleterious metabolic consequences. This is consistent with very high PRL levels ($> 100$ μg/L) observed in patients with prolactinomas and in rodent models of excessive hyperprolactinemia [46, 58, 59] being deleterious for metabolism [18, 19, 60–63] and with levels within the HomeoFIT-PRL values ($< 100$ μg/L) being metabolically beneficial.

Sulpiride is used as an antipsychotic in doses equal to or greater than 600 mg/day, which is associated with adverse metabolic effects such as weight gain and hyperglycemia [28]. On the other hand, its use as a prokinetic requires low doses ($\leq 50$ mg/day) with no metabolic effects [30]. In mice, the 30 mg/kg dose is considered equivalent to the low dose in humans because of its rapid metabolization rate in mice [64, 65] and the similar prolactinemia achieved in both species [66 and present study].

Sulpiride reduced hyperglycemia and improved glucose tolerance in obese mice due to increased insulin sensitivity rather than elevated insulin levels. Sulpiride administration ameliorated insulin resistance, reduced hyperinsulinemia, and lowered HOMA-IR in obese mice, whereas it did not alter these parameters in lean mice. These results indicate that sulpiride is metabolically beneficial in obesity conditions and does not trigger adverse metabolic effects in lean or control conditions. From these results, we conclude that sulpiride acts as a normoglycemic agent in obesity via the systemic increase in insulin sensitivity, the prevention of

hyperinsulinemia, and the reduction of insulin resistance. Of note, the observation that sulpiride decreases glucose levels in postprandial conditions in lean mice but does not improve insulin sensitivity in these conditions, suggests that other mechanisms independent from insulin sensitization are playing a role in the glucose lowering effects of sulpiride. We identified that one of the mechanisms involved in the effects of sulpiride reducing glucose levels is the increase of glycogen stores in the skeletal muscle. Dopamine is known to increase glucose uptake in the skeletal muscle and this action is mediated by D1 receptors [67]. Thus, the mechanism by which sulpiride increases glycogen stores in the skeletal muscle may be associated to the occupation and blockage of D2 receptors inducing increased availability of dopamine for D1 receptors. Interestingly, the effect of sulpiride increasing muscle glycogen levels is seen in conditions of obesity but not in lean mice, suggesting again that there are other mechanisms that account for sulpiride effects reducing glucose levels in lean animals.

Hyperprolactinemia is associated with an adverse lipid profile in patients with prolactinomas [58, 68]. In our experimental model, the sulpiride-induced increase in PRL to levels above 25 μg/L did not alter systemic total cholesterol, FFA, or glycerol in lean mice. This is consistent with the fact that PRL levels in these mice are within the HomeoFIT-PRL range (25 to 100 μg/L), which is metabolically beneficial [18, 19]. In obese mice, sulpiride reduced serum triglyceride levels. However, the HFD-induced ectopic accumulation of TG in the liver (Fig 3B) was not modified by sulpiride, indicating that sulpiride's systemic metabolic improvement does not involve a decrease in hepatic fat accumulation. However, sulpiride reduced triglyceride levels in the pancreas of obese mice, an effect that could be related to a direct interaction with the dopamine D2 receptor in this organ or to the action of elevated PRL levels on pancreatic beta cells. In support of the latter, *Prlr* is expressed in beta cells, where PRL stimulates proliferation, survival, and insulin production [69–72]. However, the effect of PRL on lipid metabolism in the pancreas has not been evaluated.

VAT dysfunction is the main trigger of metabolic alterations in obesity and is associated with adipocyte hypertrophy [7, 13, 25]. In support of its beneficial action, sulpiride prevented the greater adipocyte hypertrophy found in obese mice compared to lean mice. Moreover, adipocyte numbers were not different between obese and lean mice treated or not with sulpiride. These findings are in line with a previous study showing that elevated PRL levels reduce adipocyte hypertrophy. Ruiz-Herrera et al. reported that obese rats treated with PRL have increased VAT weight but show decreased adipocyte hypertrophy and increased adipocyte numbers [13]. Ponce et al. also showed that PRL circulating levels inversely correlate with VAT adipocyte size in humans, and low PRL levels can predict adipocyte hypertrophy [25]. The sulpiride-induced reduction of adipocyte hypertrophy is associated with a normalization in the expression of genes that are markers of adipose tissue functionality (*Insr*, *Glut4*, and *Hif1a)* [73, 74]. However, different parameters of adipocyte function (glucose uptake, lipogenesis, lipolysis, vascularization, and inflammation) require assessment to determine whether reduced hypertrophy in response to sulpiride translates into enhanced adipocyte function.

PRL involvement in sulpiride's beneficial effect on VAT is suggested by the observation that the obesogenic diet led to reduced *Prlr* expression in this tissue, which was prevented by sulpiride treatment. These findings agree with a previous report showing that subjects with insulin resistance have lower expression of *Prlr* in human VAT compared to insulin-sensitive subjects [25] and that PRL can upregulate the expression of its receptor in some tissues [75]. Therefore, *Prlr* and PRL levels are likely markers of adipose tissue functionality.

The expansion by hyperplasia of the SAT is associated with healthy tissue growth [3, 5]. Ruiz-Herrera and cols. reported that PRL treatment increases the weight and number of adipocytes (without altering their size) of the SAT in obese male rats [13]. In this work, sulpiride, despite increasing PRL levels, did not alter the weight of the SAT, however it did decrease

adipocyte hypertrophy with a tendency to increase the number of cells. The decrease in adipocyte hypertrophy suggests an improvement in the SAT metabolism in obese mice. However, unlike the effect observed in VAT, sulpiride did not cause alterations in the expression of *Insr*, *Glut4* and *Hif1α*, being *Prlr* and *Glut4* the only genes downregulated in obesity, indicating that the effects of sulpiride on SAT may be mediated through different mechanisms. Whether PRL is involved on the actions of sulpiride reducing triglyceride levels in the serum and its accumulation in the pancreas, and reducing adipocyte hypertrophy warrants further investigation.

The set of data presented here indicates that chronic treatment (30 days) with sulpiride improves metabolic homeostasis in conditions of obesity, without producing adverse effects in healthy settings, via an enhanced VAT and SAT metabolic profile. It is still necessary to evaluate whether the effect of sulpiride is mediated by elevating PRL levels or by acting directly on D2 receptors at on one or several of the organs that regulate metabolic homeostasis, such as adipose tissue, liver, pancreas, muscle, and intestine. Future experiments will allow us to learn more about the mechanisms of action of sulpiride in obesity conditions. However, our study supports a low dose of sulpiride as a potential treatment for obesity-induced metabolic dysfunction.

## Limitations

Limitations of this work include that only a single dose of sulpiride was tested on its metabolic effects in obesity, that sulpiride effects were evaluated in a single animal model of obesity, and only on male mice.

## Supporting information

**S1 Fig. Sulpiride does not affect glucagon levels in mice.** Glucagon levels were measured in serum after 30 days of sulpiride treatment using a mouse ELISA kit after 4 h of fasting. White circle: vehicle treatment; black rhombus: sulpiride treatment.
(PDF)

**S2 Fig. Sulpiride does not change fat tissue mass and limb muscle mass in mice.** A) Visceral and B) subcutaneous adipose tissue weight, and C) foreleg and hind leg muscle weight, normalized to body weight obtained from volumetric measures of body image by MRI after 30 days of sulpiride treatment. D) Images acquired with a 7.0 T MRI scanner of the coronal body of lean and obese mice after 30 days of sulpiride treatment. Groups with 2 animals were not included on the statistical analysis. White circles = vehicle treatment. Black rhombus = sulpiride treatment. **** $p < 0.0001$.
(PDF)

**S3 Fig. Sulpiride does not reduce respiratory quotient or energy expenditure.** A) Respiratory quotient (RQ) and B) energy expenditure (EE) were measured after 30 days of sulpiride treatment by indirect calorimetry in metabolic cages for 24 h. White circle: vehicle treatment; black rhombus: sulpiride treatment. ** $p < 0.01$.
(PDF)

## Acknowledgments

We thank Xarubet Ruiz-Herrera, Fernando López-Barrera, Daniel Mondragon, Antonio Prado, Martín García-Servín, Alejandra Castilla, María Antonieta Carbajo, Deisy Gasca and Juan J. Ortíz (all from Instituto de Neurobiología, UNAM) for excellent technical assistance, and Jessica González Norris (Instituto de Neurobiología, UNAM) for critically editing the

manuscript. Dina I. Vázquez Carrillo is a doctoral student from Programa de Doctorado en Ciencias Biomédicas, Universidad Nacional Autónoma de México (UNAM) and received CONAHCYT fellowship 531683.

## Author Contributions

**Conceptualization:** Dina I. Vázquez-Carrillo, Yazmín Macotela.

**Data curation:** Dina I. Vázquez-Carrillo.

**Formal analysis:** Dina I. Vázquez-Carrillo.

**Funding acquisition:** Gonzalo Martínez de la Escalera, Carmen Clapp, Yazmín Macotela.

**Investigation:** Dina I. Vázquez-Carrillo, Yazmín Macotela.

**Methodology:** Dina I. Vázquez-Carrillo, Ana Luisa Ocampo-Ruiz, Arelí Báez-Meza, Gabriela Ramírez- Hernández, Elva Adán-Castro, José Fernando García-Rodrigo, José Luis Dena-Beltrán, Ericka A. de los Ríos, Magdalena Karina Sánchez-Martínez, María Georgina Ortiz, Yazmín Macotela.

**Project administration:** Yazmín Macotela.

**Resources:** Gonzalo Martínez de la Escalera, Carmen Clapp, Yazmín Macotela.

**Supervision:** Gonzalo Martínez de la Escalera, Carmen Clapp, Yazmín Macotela.

**Validation:** Yazmín Macotela.

**Visualization:** Yazmín Macotela.

**Writing – original draft:** Dina I. Vázquez-Carrillo.

**Writing – review & editing:** Dina I. Vázquez-Carrillo, Gonzalo Martínez de la Escalera, Carmen Clapp, Yazmín Macotela.

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
