## [Decision Letter · Decision Letter 0]

29 Nov 2023

PONE-D-23-33342Dopamine D2 receptor antagonist counteracts hyperglycemia and insulin resistance in diet-induced obese male micePLOS ONE

Dear Dr. Macotela,

Thank you for submitting your manuscript to PLOS ONE. After careful consideration, we feel that it has merit but does not fully meet PLOS ONE’s publication criteria as it currently stands. Therefore, we invite you to submit a revised version of the manuscript that addresses the points raised during the review process.

Your manuscript has been carefully evaluated by three external reviewers with expertise in this field. They raised several concerns below. Especially, they suggest that the effects of sulpiride on glucose metabolism in the liver and skeletal muscles, the effects of sulpiride on prolactin secretion and levels of other glucose regulating hormone should be analyzed. These points should be addressed for further consideration. Because conclusions are not presented in an appropriate fashion and are not supported by the data, this manuscript cannot be recommended for publication in PLoS ONE in its current form.  

We look forward to receiving your revised manuscript.

Kind regards,

Wataru Nishimura, M.D., Ph.D.

Academic Editor

PLOS ONE

Journal Requirements:

4. We notice that your supplementary [Supplementary Figures 1 and 2] are included in the manuscript file. Please remove them and upload them with the file type 'Supporting Information'. Please ensure that each Supporting Information file has a legend listed in the manuscript after the references list.

Reviewers' comments:

Reviewer's Responses to Questions

**Comments to the Author**

1. Is the manuscript technically sound, and do the data support the conclusions?

Reviewer #1: Partly

Reviewer #2: Partly

Reviewer #3: Partly

2. Has the statistical analysis been performed appropriately and rigorously? 

Reviewer #1: Yes

Reviewer #2: Yes

Reviewer #3: Yes

3. Have the authors made all data underlying the findings in their manuscript fully available?

Reviewer #1: Yes

Reviewer #2: Yes

Reviewer #3: No

4. Is the manuscript presented in an intelligible fashion and written in standard English?

Reviewer #1: Yes

Reviewer #2: Yes

Reviewer #3: Yes

5. Review Comments to the Author

Reviewer #1: (Major)

As peripheral dopamine acts on Liver and skeletal muscle and adipose tissue to increases glucose uptake (Front Pharmacol. 12: 713418, 2021), blockade of D2R possibly affect glucose metabolisms in liver and muscle. At least the effects of sulpiride on glucose metabolism in the liver and skeletal muscles of lean and obese animal (glycogen content in the liver and skeletal muscles, and hepatic gluconeogenesis) should be examined.

In Fig. 2, treatment with sulpiride lowered fasting glucose levels in lean mice as well as fasting and postprandial glucose levels in obese mice. This drug showed improved insulin sensitivity in obese mice but not in lean ones. The mechanism by which sulpiride lowers fasting blood glucose cannot be explained by improving insulin sensitivity alone. Other glucose regulating hormones, such as glucagon and adrenaline, are reportedly regulated by dopamine and PRL, and measurement of these hormone levels is necessary in this study.

Although the authors conclude that sulpiride improves glucose intolerance in HFD mice via PRL secretion, the data do not demonstrate involvement of PRL and relationships between D2R inhibition-induced reduction of insulin resistance and PRL release is ambiguous. As state in discussion in the text, this effect could be rather due to the antagonism of the dopamine D2 receptors in metabolic tissues independently of PRL activity. It is interesting to examine whether the effects of sulpiride are suppressed by inhibition of PRL.

(Minor)

Fig. 3: Liver and pancreas weights are shown, but these data are not described in the text, which should be included.

Line 504: downregulated in obesity. Indicating that ... → downregulated in obesity, indicating that ...

Reviewer #2: The authors observed that hyperprolactinemia, occurring within Homo FIT-PRL levels due to the administration of the antipsychotic drug sulpiride, averted lipotoxicity and adipose tissue dysfunction while improving glucose metabolism in obese mice. This study concludes that the elevation in prolactin levels induced by sulpiride plays a role in ameliorating glucose metabolism in obese mice. However, the lack of data regarding the potential impact of D2 receptor blockade in peripheral tissues, it is questionable whether the increase in prolactin levels alone can be attributed to the improvement in glucose metabolism by sulpiride.

While the content of the manuscript is of interest to the reader, there are some points that should be clarified.

1. Roix et al. reported that intraperitoneal administration of amisulpride, a structural analog of sulpiride, elevated blood prolactin levels and enhanced glucose metabolism in DIO micehttps://doi.org/10.1111/j.1463-1326.2011.01529.x. It is crucial to elucidate the distinctive contributions of this study in relation to these reported findings.

2. While the authors describe the production of prolactin by sulpiride as a D2 receptor blocking effect in the pituitary gland, they also state in the Discussion that low concentrations of sulpiride do not cross the blood-brain barrier (p21 lines 414-420). If the intraperitoneal administration of low concentrations of sulpiride used in this experiment does not cross the blood-brain barrier, how is the increase in prolactin induced?

3. What are the effects on glucose metabolism of D2 blockers such as domperidone, which is thought to have less effect on the central nervous system than sulpiride?

Reviewer #3: The authors of this manuscript examined the effect of sulpiride on metabolic disturbances in high fat diet induced obese male mice. This manuscript has some merit, but there are some serious concerns. My comments are below:

The authors concluded that sulpiride improves metabolic disturbances including hyperglycemia and insulin resistance through increase in prolactin. However, there is no evidence showing that increase in prolactin is involved in the effect of sulpiride. Since dopamine D2 receptors widely distribute in both the brain and peripheral tissues, it is likely that sulpiride acts on the receptors in several tissues to improve metabolic disturbances. In fact, there are results showing that sulpiride showed no effect.

To show that prolactin is involved in the effect of sulpiride, it should be involved that prolactin receptor antagonist or prolactin receptor knockout counteracts the effect of sulpiride.

The authors indicate that 30 mg/kg of sulpiride “moderately” increases serum prolactin. However, since higher dose (40 mg/kg) of sulpiride shows the same level of increase in prolactin as 30 mg/kg of sulpiride, it is likely that 30 mg/kg of sulpiride occurs maximum increase in prolactin.

Data availability section should be involved.

6. PLOS authors have the option to publish the peer review history of their article (what does this mean?). If published, this will include your full peer review and any attached files.

Reviewer #1: No

Reviewer #2: No

Reviewer #3: No

---

## [Author Response · Author response to Decision Letter 0]

29 Feb 2024

Dear Editor,

We have extensively reviewed our paper to address all the reviewers concerns and comments. We have performed additional experiments to overcome the previous limitations of the study. We thank you and the reviewers for the careful revision of our work and for all the requests and suggestions that have made our work more compelling and have strengthen the message and the conclusions.

We have answered to all the requests from the Journal and from the reviewers:

Journal Requirements:

Done.

Done, we have included this text in the cover letter to the Editor: Regarding access to the data, we agree that is fully accessible and would like to send it as supporting information if the paper is accepted for publication. Thanks.

Done.

4. We notice that your supplementary [Supplementary Figures 1 and 2] are included in the manuscript file. Please remove them and upload them with the file type 'Supporting Information'. Please ensure that each Supporting Information file has a legend listed in the manuscript after the references list.

Done.

Done.

Comments to the Author

1. Is the manuscript technically sound, and do the data support the conclusions?

Reviewer #1: Partly

Reviewer #2: Partly

Reviewer #3: Partly

1. Response from authors:

We have performed new experiments that have strengthen the findings and the conclusions of the manuscript.

2. Has the statistical analysis been performed appropriately and rigorously?

Reviewer #1: Yes

Reviewer #2: Yes

Reviewer #3: Yes

3. Have the authors made all data underlying the findings in their manuscript fully available?

Reviewer #1: Yes

Reviewer #2: Yes

Reviewer #3: No

3. Response from authors

All data included on the paper is fully available. This statement has been incorporated on the manuscript, and we will submit the data as supporting information if the paper is accepted for publication.

4. Is the manuscript presented in an intelligible fashion and written in standard English?

Reviewer #1: Yes

Reviewer #2: Yes

Reviewer #3: Yes

5. Review Comments to the Author

Reviewer #1: (Major)

As peripheral dopamine acts on Liver and skeletal muscle and adipose tissue to increases glucose uptake (Front Pharmacol. 12: 713418, 2021), blockade of D2R possibly affect glucose metabolisms in liver and muscle. At least the effects of sulpiride on glucose metabolism in the liver and skeletal muscles of lean and obese animal (glycogen content in the liver and skeletal muscles, and hepatic gluconeogenesis) should be examined.

Response from authors: 

We thank the Reviewer for the critical revision of our manuscript. We agree that the effects of sulpiride can be explained by actions on glucose metabolism on liver and skeletal muscle. As requested by the reviewer, we evaluated glycogen levels on liver and skeletal muscle, and gene expression of factors involved in gluconeogenesis in the liver from lean and obese animals with and without sulpiride treatment. Interestingly, we observed that sulpiride did not alter glycogen levels or the expression of gluconeogenesis enzymes in liver, however, it increased glycogen levels in the skeletal muscle of obese mice. This result sheds light into one of the mechanisms by which sulpiride reduces glycemia in obese mice, stimulating glycogen synthesis and storage probably due to increased glucose uptake in skeletal muscle. These data have been included in the new version of the manuscript (new Figure 6). Abstract page 2, line 37, Methods page 7, lines 145-148, Results page 16, lines 368-379, and Discussion pages 21-22, lines 479-509. 

In Fig. 2, treatment with sulpiride lowered fasting glucose levels in lean mice as well as fasting and postprandial glucose levels in obese mice. This drug showed improved insulin sensitivity in obese mice but not in lean ones. The mechanism by which sulpiride lowers fasting blood glucose cannot be explained by improving insulin sensitivity alone. Other glucose regulating hormones, such as glucagon and adrenaline, are reportedly regulated by dopamine and PRL, and measurement of these hormone levels is necessary in this study.

Response from authors: 

We agree with the reviewer that the effects of sulpiride reducing glucose levels cannot be explained only by improvements of insulin sensitivity. As the reviewer points out the involvement of other mechanisms are supported by the fact that sulpiride decreases glucose levels in obese and lean animals in postprandial conditions, however, it only improves insulin sensitivity in obese animals. Following the reviewer´s request, we evaluated glucagon levels in lean and obese mice and observed that the levels of this hormone are not altered by sulpiride treatment. This data is included in the new version of the manuscript. Page 11, lines 244-246, and S1 Fig.

Although the authors conclude that sulpiride improves glucose intolerance in HFD mice via PRL secretion, the data do not demonstrate involvement of PRL and relationships between D2R inhibition-induced reduction of insulin resistance and PRL release is ambiguous. As state in discussion in the text, this effect could be rather due to the antagonism of the dopamine D2 receptors in metabolic tissues independently of PRL activity. It is interesting to examine whether the effects of sulpiride are suppressed by inhibition of PRL.

Response from authors: 

As pointed out by the reviewer, it was not clear whether the effects of sulpiride were due to increased prolactin activity derived from elevated PRL levels induced by D2R-inhibition, and this was an important limitation of the study. To address this, we have used prolactin receptor knockout mice (Prlr-/-) and treated them with sulpiride, to evaluate whether the effects of sulpiride are dependent on prolactin actions. Interestingly and contrary to our hypothesis, we found that as occurs in obese wild type mice (Prlr+/+), treatment with sulpiride resulted in reduced glucose levels in obese Prlr null mice (Prlr-/-). This demonstrates that the actions of sulpiride reducing glucose levels in obesity conditions are independent from prolactin actions, at least independent from the classical prolactin receptor mediated actions. This result has been included as new Figure 7 and incorporated throughout the paper, more extensively in the results and discussion sections of the new version of the manuscript. Page 2, lines 37-40, Page 4, lines 91-92, Page 5 lines 102-105, Page 17, lines 389-401, page 18, lines 424-428, page 19, lines 439-444, Page 24, lines 555-557.

(Minor)

Fig. 3: Liver and pancreas weights are shown, but these data are not described in the text, which should be included.

We thank the reviewer for pointing out that the description of these data was missing. It has been added to the new version of the manuscript. Page 12, lines 274-275.

Line 504: downregulated in obesity. Indicating that ... → downregulated in obesity, indicating that ...

Done, thanks. Page 24, line 554.

Reviewer #2: The authors observed that hyperprolactinemia, occurring within Homo FIT-PRL levels due to the administration of the antipsychotic drug sulpiride, averted lipotoxicity and adipose tissue dysfunction while improving glucose metabolism in obese mice. This study concludes that the elevation in prolactin levels induced by sulpiride plays a role in ameliorating glucose metabolism in obese mice. However, the lack of data regarding the potential impact of D2 receptor blockade in peripheral tissues, it is questionable whether the increase in prolactin levels alone can be attributed to the improvement in glucose metabolism by sulpiride.

While the content of the manuscript is of interest to the reader, there are some points that should be clarified.

Response from authors:

We thank the reviewer for the critical revision of our manuscript. Following the reviewer´s comments and questions, we have performed additional experiments that help clarify the study and strengthen the message and the conclusions of the study.

1. Roix et al. reported that intraperitoneal administration of amisulpride, a structural analog of sulpiride, elevated blood prolactin levels and enhanced glucose metabolism in DIO micehttps://doi.org/10.1111/j.1463-1326.2011.01529.x. It is crucial to elucidate the distinctive contributions of this study in relation to these reported findings.

Response from authors: 

We have evaluated to what extent the effects of sulpiride depend on prolactin actions. Using prolactin receptor knockout mice treated with sulpiride we observed that sulpiride reduces glucose levels in obese mice from both wild type and Prlr knockout genetic backgrounds, demonstrating that the effects of sulpiride reducing glucose levels are independent of prolactin receptor activity. It is still possible that other effects of sulpiride involve prolactin action. 

Moreover, we evaluated whether the effects of sulpiride involve increased glycogen stores in liver and skeletal muscle. We observed that sulpiride treatment elevates skeletal muscle glycogen stores. This result constitutes novel information around the mechanisms that sulpiride uses to reduce glucose levels in obesity conditions. These two new pieces of data clearly differentiate the contributions of this paper from the paper by Roix et al., 2011. Also, in contrast to the findings of Roix et al., we found that the effects of sulpiride do not involve increased insulin levels.

The new data has been incorporated in new figures 6 and 7 as well as in the results and discussion sections of the paper. Page 2, lines 37-40, Page 4, lines 91-92, Page 5 lines 102-105, Page 7, lines 145-148, Page 16, lines 368-379, Page 17, lines 389-401, pages 18-19, lines 424-428, page 19, lines 439-444, Pages 21-22, lines 497-509, Page 24, lines 555-557.

2. While the authors describe the production of prolactin by sulpiride as a D2 receptor blocking effect in the pituitary gland, they also state in the Discussion that low concentrations of sulpiride do not cross the blood-brain barrier (p21 lines 414-420). If the intraperitoneal administration of low concentrations of sulpiride used in this experiment does not cross the blood-brain barrier, how is the increase in prolactin induced?

The pituitary gland lies outside the blood brain barrier, therefore, sulpiride injected intraperitoneally can easily reach their D2 receptors on the pituitary gland lactotrophs, therefore antagonizing dopamine action on these receptors.

3. What are the effects on glucose metabolism of D2 blockers such as domperidone, which is thought to have less effect on the central nervous system than sulpiride?

Tavares et al, 2021, showed that domperidone abolishes the effect of dopamine and insulin stimulating glucose uptake in liver. However, dopamine alone or in combination with domperidone did not alter liver glucose uptake, suggesting that D2R plays a role on insulin-induced liver glucose uptake. This effect was further confirmed by using bromocriptine, that in combination with insulin increased further glucose uptake, and the effect was reduced by domperidone. Domperidone also reduced the effect of insulin + dopamine, and insulin + bromocriptine stimulating glucose uptake in epididymal white adipose tissue, supporting that also in adipose tissue D2R are involved in the effects of insulin stimulating glucose uptake. In skeletal muscle, dopamine was shown to increase glucose uptake via D1 receptors (Tavares et al., 2021), as the effect was not elicited by bromocriptine (D2R agonist) and was blocked by haloperidol (D1R and D2R antagonist). However, the effect of domperidone on muscle glucose uptake was not tested. Thus, the mechanism by which sulpiride increases glycogen stores in the skeletal muscle may be associated to the occupation and blockage of D2 receptors inducing increased availability of dopamine for D1 receptors. This has been included in the discussion of the paper. Pages 21-22, lines 497-509.

Reviewer #3: The authors of this manuscript examined the effect of sulpiride on metabolic disturbances in high fat diet induced obese male mice. This manuscript has some merit, but there are some serious concerns. My comments are below:

The authors concluded that sulpiride improves metabolic disturbances including hyperglycemia and insulin resistance through increase in prolactin. However, there is no evidence showing that increase in prolactin is involved in the effect of sulpiride. Since dopamine D2 receptors widely distribute in both the brain and peripheral tissues, it is likely that sulpiride acts on the receptors in several tissues to improve metabolic disturbances. In fact, there are results showing that sulpiride showed no effect.

To show that prolactin is involved in the effect of sulpiride, it should be involved that prolactin receptor antagonist or prolactin receptor knockout counteracts the effect of sulpiride.

Response from authors:

We appreciate and thank the reviewer for the critical revision of our work. As the reviewer points out, the paper lacked the evidence regarding the involvement of prolactin on the effects of sulpiride decreasing glucose levels in obese mice. To overcome this limitation and in response to the reviewer´s request, we have used prolactin receptor knockout mice to test whether prolactin actions are necessary for the effects of sulpiride. Interestingly and contrary to our hypothesis we found that as occurs in obese wild type mice, supiride exerts its glucose-reducing action also in obese prolactin receptor (Prlr) null mice. Moreover, the effect of sulpiride appears to be of the same magnitude in both wild type and Prlr null mice, demonstrating that at least for the effects of sulpiride on glucose levels, the activity of prolactin/prolactin receptor is not required. This result has been included as new Figure 7 and incorporated in the results and discussion sections of the new version of the manuscript. Page 2, lines 37-40, Page 4, lines 91-92, Page 5 lines 102-105, Page 17, lines 389-401, page 18, lines 424-428, page 19, lines 439-444, Page 24, lines 555-557.

The authors indicate that 30 mg/kg of sulpiride “moderately” increases serum prolactin. However, since higher dose (40 mg/kg) of sulpiride shows the same level of increase in prolactin as 30 mg/kg of sulpiride, it is likely that 30 mg/kg of sulpiride occurs maximum increase in prolactin.

Response from authors:

The reviewer is right that the dose of 30 mg/kg of sulpiride achieved a maximal PRL level as PRL levels did not increase further with 40 mg/kg. We used the word “moderately” to compare this level to a high hyperprolactinemia level, usually considered when PRL levels are above 100 ng/mL.

Data availability section should be involved.

Response from authors:

In agreement with the reviewer´s request, we have included a legend stating that data is fully available. 

6. PLOS authors have the option to publish the peer review history of their article (what does this mean?). If published, this will include your full peer review and any attached files.

Do you want your identity to be public for this peer review? For information about this choice, including consent withdrawal, please see our Privacy Policy.

Reviewer #1: No

Reviewer #2: No

Reviewer #3: No

---

## [Decision Letter · Decision Letter 1]

15 Mar 2024

PONE-D-23-33342R1Dopamine D2 receptor antagonist counteracts hyperglycemia and insulin resistance in diet-induced obese male micePLOS ONE

Dear Dr. Macotela,

Thank you for submitting your manuscript to PLOS ONE. After careful consideration, we feel that it has merit but does not fully meet PLOS ONE’s publication criteria as it currently stands. Therefore, we invite you to submit a revised version of the manuscript that addresses the points raised during the review process.

Your manuscript has been carefully evaluated by two external reviewers with expertise in this field. One of the reviewers suggested that authors should describe the genetic background and origin of prolactin receptor KO mice in the methods. Please address this comment for publication.  

We look forward to receiving your revised manuscript.

Kind regards,

Wataru Nishimura, M.D., Ph.D.

Academic Editor

PLOS ONE

Journal Requirements:

Reviewers' comments:

Reviewer's Responses to Questions

**Comments to the Author**

1. If the authors have adequately addressed your comments raised in a previous round of review and you feel that this manuscript is now acceptable for publication, you may indicate that here to bypass the “Comments to the Author” section, enter your conflict of interest statement in the “Confidential to Editor” section, and submit your "Accept" recommendation.

Reviewer #1: (No Response)

Reviewer #2: (No Response)

2. Is the manuscript technically sound, and do the data support the conclusions?

Reviewer #1: Yes

Reviewer #2: (No Response)

3. Has the statistical analysis been performed appropriately and rigorously? 

Reviewer #1: Yes

Reviewer #2: (No Response)

4. Have the authors made all data underlying the findings in their manuscript fully available?

Reviewer #1: Yes

Reviewer #2: (No Response)

5. Is the manuscript presented in an intelligible fashion and written in standard English?

Reviewer #1: Yes

Reviewer #2: (No Response)

6. Review Comments to the Author

Reviewer #1: Regarding prolactin receptor KO mice, please explain or cite the genetic background and mouse origin.

Reviewer #2: (No Response)

7. PLOS authors have the option to publish the peer review history of their article (what does this mean?). If published, this will include your full peer review and any attached files.

Reviewer #1: No

Reviewer #2: No

---

## [Author Response · Author response to Decision Letter 1]

15 Mar 2024

Dear Editor,

We have addressed the request from the reviewer regarding more detailed information about the prolactin receptor null mice. This information has been included in the Methods section of the paper. Page 5, lines 106-109 including one new reference (new reference 33). Also, two references were changed (references 29 and 49): one reference was changed because the DOI was mismatched with the reference, the correct reference was added. The other reference was removed as the DOI did not lead to the paper and the journal page does not have the paper anymore. The reference numbers were adjusted accordingly throughout the paper.

Journal Requirements:

Authors response: We have carefully reviewed the reference list and did not detect any retracted paper in our reference list. However, two references were removed, one because the DOI was not matching the reference, therefore the correct reference was added to match the DOI as follows:

Old reference:

29. Mansi C, et al. (1995). Gastrokinetic effects of levosulpiride in dyspeptic patients with diabetic gastroparesis. Am J Gastroenterol 90(11) 1989-93. 

New reference:

29. Mansi C, et al. (2000). Comparative effects of levosulpiride and cisapride on gastric emptying and symptoms in patients with functional dyspepsia and gastroparesis. Aliment Pharmacol Ther 14, 561-569. https://doi.org/10.1046/j.1365-2036.2000.00742.x

And the other one, reference 49 was removed as the DOI was not working and the journal did not have access to the paper (maybe this is the retracted reference?):

49. Durant, S., Coulaud, J., & Homo-Delarche, F. (2007). Bromocriptine-induced hyperglycemia in nonobese diabetic mice: Kinetics and mechanisms of action. Review of Diabetic Studies, 4(3), 185–194. https://doi.org/10.1900/RDS.2007.4.185

Reviewers' comments:

Reviewer's Responses to Questions

Comments to the Author

1. If the authors have adequately addressed your comments raised in a previous round of review and you feel that this manuscript is now acceptable for publication, you may indicate that here to bypass the “Comments to the Author” section, enter your conflict of interest statement in the “Confidential to Editor” section, and submit your "Accept" recommendation.

Reviewer #1: (No Response)

Reviewer #2: (No Response)

2. Is the manuscript technically sound, and do the data support the conclusions?

Reviewer #1: Yes

Reviewer #2: (No Response)

3. Has the statistical analysis been performed appropriately and rigorously?

Reviewer #1: Yes

Reviewer #2: (No Response)

4. Have the authors made all data underlying the findings in their manuscript fully available?

Reviewer #1: Yes

Reviewer #2: (No Response)

5. Is the manuscript presented in an intelligible fashion and written in standard English?

Reviewer #1: Yes

Reviewer #2: (No Response)

6. Review Comments to the Author

Reviewer #1: Regarding prolactin receptor KO mice, please explain or cite the genetic background and mouse origin.

Authors response: We have added detailed information about genetic background and mouse origin of the prolactin receptor KO mice, including a reference. Page 5, lines 106-109: The colony of prolactin receptor null mice has been expanded and maintained for many generations in the vivarium of our Institute and was originally started from Prlr-/+ mice on a C57BL/6 background, obtained from The Jackson Laboratory (www.jax.org; strain:003142, B6.129P2-Prlrtm1Cnp/J) [33]).

New reference 

33. Ormandy, C.J., Camus, A., Barra J., Damotte, D., Lucas, B., Buteau, H., Edery, M., Brouss, N., Babinet, C., Binart, N., Kelly P.A. (1997). Null mutation of the prolactin receptor gene produces multiple reproductive defects in the mouse. Genes & Dev. 11: 167 – 178. https://doi.org/10.1101/gad.11.2.167

Reviewer #2: (No Response)

7. PLOS authors have the option to publish the peer review history of their article (what does this mean?). If published, this will include your full peer review and any attached files.

Do you want your identity to be public for this peer review? For information about this choice, including consent withdrawal, please see our Privacy Policy.

Reviewer #1: No

Reviewer #2: No

---

## [Editor Report · Decision Letter 2]

18 Mar 2024

Dopamine D2 receptor antagonist counteracts hyperglycemia and insulin resistance in diet-induced obese male mice

PONE-D-23-33342R2

Dear Dr. Macotela,

We’re pleased to inform you that your manuscript has been judged scientifically suitable for publication and will be formally accepted for publication once it meets all outstanding technical requirements.

Kind regards,

Wataru Nishimura, M.D., Ph.D.

Academic Editor

PLOS ONE
---

## [Editor Report · Acceptance letter]

28 Mar 2024

PONE-D-23-33342R2 

PLOS ONE

Dear Dr. Macotela, 

I'm pleased to inform you that your manuscript has been deemed suitable for publication in PLOS ONE. Congratulations! Your manuscript is now being handed over to our production team.

Kind regards, 

on behalf of

Dr. Wataru Nishimura 

Academic Editor

PLOS ONE